# Diversity and Complexity of Internally Deleted Viral Genomes in Influenza A Virus Subpopulations with Enhanced Interferon-Inducing Phenotypes

**DOI:** 10.3390/v15102107

**Published:** 2023-10-17

**Authors:** Amir Ghorbani, John M. Ngunjiri, Gloria Rendon, Christopher B. Brooke, Scott P. Kenney, Chang-Won Lee

**Affiliations:** 1Department of Veterinary Preventive Medicine, College of Veterinary Medicine, The Ohio State University, Columbus, OH 43210, USA; amir.ghorbani@nih.gov; 2Center for Food Animal Health, Ohio Agricultural Research and Development Center, The Ohio State University, Wooster, OH 44691, USA; 3Carl R. Woese Institute for Genomic Biology, University of Illinois at Urbana-Champaign, Urbana, IL 61801, USAcbrooke@illinois.edu (C.B.B.); 4Department of Microbiology, University of Illinois at Urbana-Champaign, Urbana, IL 61801, USA; 5Southeast Poultry Research Laboratory, US National Poultry Research Center, USDA, ARS, Athens, GA 30605, USA

**Keywords:** influenza, subpopulations, interferon, defective genomes, NS1 truncation

## Abstract

Influenza A virus (IAV) populations harbor large subpopulations of defective-interfering particles characterized by internally deleted viral genomes. These internally deleted genomes have demonstrated the ability to suppress infectivity and boost innate immunity, rendering them promising for therapeutic and immunogenic applications. In this study, we aimed to investigate the diversity and complexity of the internally deleted IAV genomes within a panel of plaque-purified avian influenza viruses selected for their enhanced interferon-inducing phenotypes. Our findings unveiled that the abundance and diversity of internally deleted viral genomes were contingent upon the viral subculture and plaque purification processes. We observed a heightened occurrence of internally deleted genomes with distinct junctions in viral clones exhibiting enhanced interferon-inducing phenotypes, accompanied by additional truncation in the nonstructural 1 protein linker region (NS1Δ76-86). Computational analyses suggest the internally deleted IAV genomes can encode a broad range of carboxy-terminally truncated and intrinsically disordered proteins with variable lengths and amino acid composition. Further research is imperative to unravel the underlying mechanisms driving the increased diversity of internal deletions within the genomes of viral clones exhibiting enhanced interferon-inducing capacities and to explore their potential for modulating cellular processes and immunity.

## 1. Introduction

The influenza A virus (IAV) quasispecies consists of a highly diverse collection of genetically and biologically different subpopulations of viral particles [1,2,3]. These subpopulations collectively impact the virus’s evolutionary fitness. In the past, most particles in IAV populations were considered biologically insignificant since they could not be quantified using infectivity assays. However, around 75 years ago, Preben von Magnus identified IAV defective-interfering (DI) particles, which are incomplete virus particles that impede the replication of fully functional virus particles [4,5]. The IAV DI particles have since been widely observed in laboratory-grown virus cultures [6,7,8] and clinical samples [9,10]. Recently, there has been growing interest in the therapeutic and immunogenic properties of DI particles [11,12,13,14,15,16,17]. They have also been suggested to contribute significantly to the safety and effectiveness of live attenuated influenza vaccines (LAIVs) by potentially reducing infectivity and enhancing innate immunity [11,17,18,19,20].

The IAV DI particles exhibit large internal deletions within one or more genomic segments while retaining the necessary elements for polymerase binding, replication, and packaging at the 3′ and 5′ termini of the segment [21]. These deletions are most commonly found in the largest viral genomic segments that encode the polymerase basic 1 (PB1), polymerase basic 2 (PB2), and polymerase acidic (PA) proteins [7,22,23]. Due to their preferential replication compared to full-length vRNAs, these internally deleted viral RNAs (ID vRNAs) serve as one of the mechanisms through which IAV DI particles inhibit or restrict virus production in cells coinfected with fully infectious virions [7,22,23,24,25,26]. In addition to DI particle-derived ID vRNAs, which are typically larger than ~200 nucleotides, there are other classes of short aberrant vRNAs, namely mini and small vRNAs, with lengths of ~50–125 and 22–27 nucleotides, respectively, which are less likely to be encapsulated within influenza virions [27,28,29,30]. ID vRNAs play a vital role in activating the innate immune response by specifically binding to retinoic acid-inducible gene-I (RIG-I) [9,31,32], leading to increased expression of interferon (IFN) and other innate genes in cells [33]. Owing to the potent adjuvant effects of IFNs, ID vRNAs hold great potential as targets for developing LAIVs that can elicit enhanced innate and adaptive immune responses [34,35,36,37,38].

We have previously introduced the plaque clone 4 (PC4) virus, with a severely truncated nonstructural 1 (NS1) protein effector domain but intact RNA-binding and linker domains, that shows promising efficacy as an avian LAIV. The NS1 protein serves a pivotal role in antagonizing the host’s IFN response, and its truncation in IAVs results in enhanced innate immune responses, making it a promising candidate for the development of highly attenuated and effective LAIVs [39,40]. Accordingly, the PC4 virus elicits strong innate immune responses in avian cell cultures and the tracheas of vaccinated chickens [17,38,41,42,43,44,45] and provides broad protection against both heterologous and heterosubtypic IAV challenges in vaccinated animals [17,38,41,42,43,44,45]. Importantly, these attributes are consistent with the abundance of subpopulations of DI particles in the PC4 virus preparation [17]. To better understand the diversity within the IAV quasispecies, we analyzed the IFN-inducing capabilities of a large collection of plaque-purified PC4 virus clones. Among these clones, a small subset exhibited an additional deletion in their NS1 protein linker domain (NS1Δ76-86) and induced significantly enhanced IFN responses in cell cultures compared to the majority of viral subpopulations with similar genotypes/phenotypes (referred to as parental clones) [44]. However, the diversity and complexity of DI particles or their ID vRNAs within plaque-purified viral subpopulations with distinct IFN-inducing capacities have yet to be elucidated.

Deep sequencing and high-performance supercomputing techniques have allowed us to identify single nucleotide changes in viral genomes and characterize the DI particle-derived ID vRNAs within viral populations [8,46]. In this study, we investigated the abundance and diversity of ID vRNAs in the PC4 virus and several of its plaque-purified clones with parental or high IFN-inducing phenotypes carrying the NS1Δ76-86 mutation. We first demonstrated the varying effect of virus subculture or plaque purification on the abundance and diversity of ID vRNAs within IAV populations. We then examined the diversity of ID vRNA junctions within the parental and high IFN-inducing clones. Interestingly, the high IFN-inducing clones generated larger number of ID vRNAs with distinct junctions across their genome than the parental clones. Overall, our findings highlight the widespread occurrence of ID vRNAs with distinct junctions in IAV populations, which can potentially encode a broad range of non-canonical proteins with variable length and amino acid composition.

## 2. Materials and Methods

### 2.1. Viruses and Cells

The A/turkey/OR/71 (H7N3) virus strain with a truncated NS1 protein (PC4 P1) was de novo generated by using reverse genetics [41] and then cultured in 10-day-old specific-pathogen-free embryonated chicken eggs from an in-house chicken flock at The Ohio State University in Wooster, OH. To prepare the 2nd PC4 passage (PC4 P2), a 1:10,000 dilution of the original PC4 virus was subcultured in 10-day-old embryonated chicken eggs [44]. Clonal virus populations were obtained by plaque purifying them from freshly prepared chicken embryo fibroblast cells and subsequently propagated in 10-day-old embryonated chicken eggs, following previously established protocols [44]. Virus infectivity was titrated using plaque assay in freshly prepared embryo kidney cells with no exogenous trypsin [44].

Chicken embryo fibroblast and kidney cells were derived from 10- and 18-day-old embryonated chicken eggs, respectively [47]. The primary cells were grown in Roswell Park Memorial Institute (RPMI) 1640 medium (Gibco, catalog number 11875135) containing 10% fetal bovine serum (FBS) (Gibco, catalog number 16000044), 10 μg/mL gentamicin (Gibco, catalog number 15710072) and 0.25 µg/mL amphotericin B (Gibco, catalog number 15290018) at 39 °C and 5% CO_2_ in a humidified cell culture incubator. Chicken embryo fibroblast cells were allowed to continue to grow for 10 days in RPMI medium supplemented with 5% FBS, 10 μg/mL gentamicin and 0.25 µg/mL amphotericin B at 39 °C and 5% CO_2_ to prepare the developmentally aged chicken embryo fibroblasts [17,48,49,50]. QT-35 cells (ECACC, catalog number 93120832) were cultivated in minimum essential medium (MEM) (Gibco, catalog number 11095080) with 10% FBS and 10 μg/mL gentamicin at 37 °C and 5% CO_2_ in a humidified cell culture incubator.

### 2.2. Genome Amplification, Deep Sequencing, and Sequence Analysis

Viral RNA extraction was performed using the QIAamp viral RNA mini kit (Qiagen, catalog number 52906) according to the manufacturer’s instructions, from 100 µL of virus stocks. Duplicates of viral RNA were extracted from each of the PC4 stocks (P1 and P2) to be included for sequencing to examine the reproducibility of the results. Before deep sequencing, the amplification of the IAV genome was conducted following established protocols [46,51]. In brief, viral RNA was reverse transcribed into cDNA using SuperScript IV Reverse Transcriptase (Invitrogen, catalog number 18090050) and the MBTUni-12 primer (5′-ACGCGTGATCAGCAAAAGCAGG-3′). The cDNA was prediluted to contain 4 × 10^6^ genome copies (based on IAV matrix gene), underwent PCR amplification using the Phusion High-Fidelity DNA Polymerase (NEB, catalog number M0530S) with the MBTUni-13 (5′-ACGCGTGATCAGTAGAAACAAGG-3′) and MBTUni-12 (5′-ACGCGTGATCAGCAAAAGCAGG-3′) primers. The PCR amplification followed a thermal program consisting of an initial denaturation at 98 °C for 30 s followed by 25 cycles of denaturation at 98 °C for 10 s, annealing at 57 °C for 30 s, and extension at 72 °C for 90 s, and a final extension at 72 °C for 5 min. The reaction was then held at 10 °C [44]. To visualize the whole genomic profiles of selected samples, amplified PCR products were loaded on a 2% *w*/*v* EtBr-stained agarose gel and subjected to gel electrophoresis in Tris-acetate-EDTA buffer for 105 min at 110 V. For deep sequencing preparation, the PCR products underwent direct purification using the PureLink PCR purification kit (Invitrogen, catalog number K310001) with the lower-cutoff option. The purified products were then utilized for library preparation using the Kapa Hyper prep kit (Roche) [44]. Deep sequencing was performed using the paired-end Illumina MiSeq platform (2 × 250 bp) at the University of Minnesota Genomics Center (Minneapolis, MN). The determination of consensus genomic sequences and coverage depths was performed using the web-based INSaFLU bioinformatics suite [52]. Analysis of ID vRNAs was performed using the modified Viral Recombination Mapper (ViReMa) algorithm (v0.10) [8,46]. Briefly, Trimmomatic (v0.36) was used to quality-filter the raw sequencing reads with the following parameters “ILLUMINACLIP:adapters/TruSeq3-PE-2.fa:2:15:10 SLIDINGWINDOW:3:20 LEADING:28 TRAILING:28 MINLEN:75”. Next, Bowtie 2 (v2.3.1) was used to concatenate the paired reads into one file with the following parameters: “-score-min L, 0, −0.3”. Subsequently, ViReMa (v0.10) was used to identify the putative break and rejoin sites within the unaligned reads with the following parameters: “-params.micro = ‘20’ -params.defuzz = ‘3’ -params.mismatch = ‘1’ -params. X = 8 -params.minCoverage = 5 -params.filterType = 1”. The percent abundance of ID vRNAs was estimated based on the percentage of “number of detected ID vRNA” divided by “the mean depth of coverage at the 5′ and 3′ termini of the genomic segments”. The mean depth of coverage at the 5′ and 3′ ends of each viral genomic segment was estimated based on the average depth of coverage for the first 100 nucleotides at each gene termini, excluding the first 20 nucleotides corresponding to the primer sequences. Only the ID vRNAs with an estimated percent abundance of ≥0.1% were included in the downstream analysis to account for sequencing and sequence analysis errors. ID vRNAs coding sequences were reconstructed based on the identified break and rejoint sites in R (v3.5.1). Putative open reading frames were identified by looking for translation initiation (‘AUG’) and termination (‘UGA’, ‘UAG’, or ‘UAA’) sites assuming a minimum length of 150 nucleotides and no reinitiation downstream of the first ‘AUG’ start codon in Python (v3.10.9). Multiple sequence alignment of the putative polypeptide sequences were performed based on the Clustal W method on slow/accurate mode and BLOSUM weight matrix on the GenomeNet (https://www.genome.jp/tools-bin/clustalw (accessed on 6 August 2023)) [53,54,55]. Phylogenetic reconstructions were performed using the function “build” of ETE3 3.1.2 [56] as implemented on the GenomeNet (https://www.genome.jp/tools/ete/ (accessed on 6 August 2023)). The tree was constructed using FastTree with slow NNI and MLACC = 3 (to make the maximum-likelihood NNIs more exhaustive) [57]. Prediction of intrinsically disordered polypeptides and protein binding regions was performed using the IUPred3 and ANCHOR2 algorithms with the default “long disorder” option [58,59,60].

### 2.3. Induction and Quantification of Biologically Active IFNs

The induction, purification, and quantification of biological activity of acid-stable type I IFNs followed our established protocol [44]. To induce the peak IFN responses, developmentally aged chicken embryo fibroblasts were infected with the indicated viruses at an MOI of 0.3. Type I IFNs released into the cell supernatants by 24 hpi were purified via a perchloric acid-based protein precipitation method [6,44,49,61]. The antiviral activity of the purified IFNs was evaluated against vesicular stomatitis virus (VSV) infection in QT-35 cell monolayer, following previously described methods [44]. Relative antiviral activities were determined by comparing the measured IFN units in each tested sample to the average IFN units induced by the PC4 virus in the same experiment.

### 2.4. Statistical Analysis, Visualization, and Availability of Data

GraphPad Prism 9 software (San Diego, CA) was utilized for statistical analysis and data visualization. Statistical significance was assessed using the unpaired *t*-test for experiments with two groups and one-way analysis of variance (ANOVA) followed by Tukey’s post hoc test for experiments with more than two groups. A *p* value of <0.05 was considered statistically significant for all analyses. Circular representation of the deletion junction sites was illustrated using a modified JavaScript code based on a publicly available visualization tool (http://www.di-tector.cyame.eu/ (accessed on 6 August 2023)) with minor modifications in object color, text size, and image resolution [62]. Phylogenetic trees, including the additional ID vRNA abundance and protein length metadata, were visualized using the Interactive Tree Of Life (iTOL) [63]. All assembled consensus sequences and raw sequence reads have been submitted to the NCBI GenBank (MW080971-MW081050) and Sequence Read Archive (PRJNA668426 and SRX9274256-SRX9274266) databases.

## 3. Results

### 3.1. ID vRNA Abundance and Pattern Can Be Preserved in Subcultured PC4 Virus

To examine the whole genomic profile of the ID vRNAs within the viral preparations, we performed a universal RT-PCR on the vRNAs extracted from the original PC4 virus synthesized de novo through reverse genetics (PC4 P1) and a subsequent passage of the virus 10 years later (PC4 P2), along with a few PC4-derived plaque clones [44]. When we subjected the amplified PCR products to gel electrophoresis, both PC4 passages showed a similar pattern for subgenomic ID vRNAs within ~250–700 bp size range just below the truncated nonstructural (NS) gene band [700 bp [41]] (Figure 1). In sharp contrast, the plaque-purified clones’ ID vRNA size profiles differed from one another and the PC4 passages. It is worth noting that the observed ID vRNAs are limited to those shorter than the truncated NS gene, the smallest genomic segment of the PC4 virus. Therefore, the complete profile of the ID vRNAs cannot be assessed through the universal RT-PCR and electrophoresis approach [46,51].

To further characterize the ID vRNAs, we deep sequenced the viral stocks in duplicate and analyzed the raw sequences through the ViReMa bioinformatics pipeline designed to identify the defective forms of IAV genomes in deep sequence data [46]. Figure 2A compares the abundances of ID vRNAs between PC4 virus passages across different viral genes estimated based on the percentage of ID vRNAs over average sequence coverages at 5′ and 3′ termini of each gene segment. These results further support two general conclusions from previous studies [7,10,22,23]. First, the majority (>90%) of the ID vRNAs are derived from the largest genomic segments encoding the trimeric polymerase proteins (PB2, PB1, and PA) (Figure 2A). Second, a sizable proportion of each IAV population can be occupied by ID vRNAs originating from one or more of the viral genomic segments. Interestingly, on average, the relative abundance of ID vRNAs per nucleotide of each viral genome segment (relative abundance divided by segment size) for PB2, PB1, and PA genes was >90 times higher than that of all the other genes in both viral stocks (Appendix A). Additionally, we observed a significantly higher abundance of the ID vRNAs from the PB2 gene than from the PB1 or PA genes in both viral stocks (Figure 2A). Figure 2B represents the number of distinct deletion junctions identified within the sequence data. Interestingly, deletion junctions found across the PB1 gene were statistically higher than those identified in the PB2 or PA genes for both viral stocks (Figure 2B). Additionally, there was a significant drop in the number of deletion junctions associated with the PB1 in the 2nd PC4 passage, which might be related to the loss of some larger-sized ID vRNAs (Figure 2B,C (e.g., length ~1000 bp), and D (e.g., break and rejoin sites at ~500 and 1800 bp, respectively). Nevertheless, the ID vRNAs abundance and size profiles were generally consistent between the two viral passages (Figure 2A,C, respectively), which correlate with the result of RT-PCR and gel electrophoresis (Figure 1). Together, these results suggest that once ID vRNAs are produced de novo, most of them can be preserved during subsequent passage of the virus in eggs. However, it is essential to note that the observed pattern may not be the same for all IAV strains and during viral passages at different multiplicities of infections or hosts.

### 3.2. Physical Separation of PC4 Viral Subpopulations through Plaque Purification Significantly Reduced ID vRNA Abundance

To investigate how physical separation affects PC4 viral subpopulations and their ID vRNA profiles, we compared the ID vRNA profiles of the PC4 P2 virus used for plaque purification with those of the five parental clones exhibiting similar IFN-inducing abilities to the parental virus as documented in our previous publication [44]. Figure 3 offers a schematic illustration of the experimental design and population hierarchy pertaining to PC4-related viral subpopulations. The parental clones possess no or minimal single nucleotide changes only in their neuraminidase (NA; D127N [clone 3]) or hemagglutinin (HA; I59V [clone 4] and V332F [clone 5]) surface glycoprotein genes and induce similar levels of IFNs in cell cultures when compared to the PC4 virus [44]. Interestingly, plaque purification significantly reduced the abundance of PB2-derived ID vRNAs in the parental clones compared to the PC4 virus (Figure 4A). Despite the reduced abundance, ID vRNAs with distinct deletion junctions emerged in almost all plaque-purified clones at similar levels to the PC4 virus (Figure 4B). Notably, all the viral clones reached similar or higher infectious titers than the PC4 virus during the initial passage in embryonated chicken eggs. Although most ID vRNA junctions occurred within similar areas at the polymerase gene termini, some levels of dissimilarity in the sizes and junction profiles were observed between parental clones versus the PC4 virus (Figure 4C,D). This is likely because most ID vRNAs identified in the plaque-purified clones are expected to have been generated de novo and, therefore, have not been subjected to a strong selective pressure caused by other ID vRNAs (e.g., smaller species, etc.). Altogether, these results indicate that although the physical separation of IAV subpopulations through plaque purification significantly affects ID vRNAs’ content, it does not prevent de novo generation of ID vRNAs.

High IFN-inducing clones possessing the NS1Δ76-86 deletion contain larger number of de novo ID vRNAs with distinct deletion junctions. To investigate the potential association between ID vRNA content and IFN-inducing capacities of plaque-purified IAV subpopulations, we compared the ID vRNA profiles of the five parental clones with those of the three high IFN-inducing clones that possess the NS1Δ76-86 deletion along with only a few single nucleotide changes in NA (F97L and E119A [NS1Δ76-86 clones 1–3] or HA (A168G [NS1Δ76-86 clone 3]) [44]. All three NS1Δ76-86 clones showed a similar abundance of ID vRNAs in each genomic segment compared to the parental clones (Figure 5A). Strikingly, ID vRNAs detected in the NS1Δ76-86 clones showed a significantly higher number of distinct deletion junctions associated with the PB2 gene than those detected in the parental clones (Figure 5B). Like our previous observation with the parental clones, most breakpoints occurred within similar areas of the polymerase genes, and therefore, no difference in the size profiles was detected (Figure 5C). Accordingly, no clear discriminative pattern was observed to differentiate the parental from the NS1Δ76-86 clones solely based on their ID vRNA sizes and break and rejoin sites. However, the NS1Δ76-86 clones appeared to possess more (not statistically significant for PB1 and PA) distinct deletion junctions within all three polymerase genes (Figure 5D, golden yellow bands).

Similarly, in principal component analysis (PCoA) based on Euclidean distances among the abundance of ID vRNA and the break and rejoin sites, the NS1Δ76-86 clones appeared to exhibit greater separation from each other compared to the parental clones (Appendix A). Next, we looked at the total abundance of ID vRNAs and the number of distinct deletion junctions detected across all genomic segments in parental versus the NS1Δ76-86 clones (Figure 5A,B). While the total abundance of ID vRNAs was not statistically different between the parental and NS1Δ76-86 clones (Figure 6A), the total number of unique deletion junctions detected from all eight genomic segments combined was also significantly higher in the NS1Δ76-86 clones than in the parental clones (Figure 6B). Altogether, these results suggest that the NS1Δ76-86 clones are more likely to generate de novo ID vRNAs with distinct break and rejoin sites compared to the parental clones (Figure 6B), which correlate with the enhanced IFN responses triggered by the NS1Δ76-86 clones in cell cultures (Figure 6C). It is still unclear whether such ID vRNAs can play any role in the higher induction of IFNs or are byproducts of higher innate immune responses induced in the infected cells.

### 3.3. ID vRNAs Can Encode Large Libraries of Intrinsically Disordered Polypeptides with Potential Protein Binding Sites

Almost all the ID vRNAs identified in this study maintain all the elements required for polymerase binding, replication, and packaging at their 3′ and 5′ termini [21]. Therefore, most ID vRNAs should be able to express truncated or frameshifted forms of the viral proteins. To investigate the diversity of potential polypeptides that could be expressed from the detected ID vRNAs, we used the break and rejoin sites identified within the polymerase genes of the PC4-derived plaque-purified clones to reconstruct the potential truncated polypeptides that could be generated from the ID vRNAs in silico. Strikingly, many identified ID vRNAs appear to contain frameshift mutations, leading to extremely high diversity in the predicted polypeptides. Figure 7 represents the phylogenetic diversity of the reconstructed polypeptides derived from the polymerase genes of the plaque-purified clones based on the maximum-likelihood clustering method [53,54,55]. Interestingly, >200 forms of viral polypeptides could be reconstructed based on the ID vRNAs identified for each polymerase gene (Figure 7). Comparatively, lower diversity appears to exist in polypeptides derived from the PB1 gene compared to the ones from PB2 and PA, as evident with fewer visible leaf nodes after collapsing all the clades with mean branch lengths of <0.1 or <0.2 in the phylogenetic tree drawn for the PB1-derived polypeptides (Figure 7). These results suggest that the ID vRNAs may express large libraries of non-canonical viral polypeptides during virus infection.

We then asked if the proposed polypeptide encodes intrinsically disordered polypeptides containing potential protein binding regions using the IUPred3 and ANCHOR2 prediction algorithms [58,59,60]. These models predict protein regions lacking a stable three-dimensional structure by assessing the propensity of each amino acid residue to be disordered (probability values from ‘0’, least likely, to ‘1’, most likely). Interestingly, most of the proposed polypeptides originating from the PB2 and PB1 ID vRNAs are more likely to be disordered than the average baseline value based on the intact PB2 or PB1 viral proteins (Figure 8). In sharp contrast, the polypeptides originating from the PA ID vRNAs are less likely to be disordered when compared to the intact PA protein (Figure 8). However, no significant differences were observed between the average IUPred3 and ANCHOR2 values among the viral subpopulations with distinct IFN-inducing phenotypes (Appendix A). Given the ubiquitous nature of ID vRNAs, the proposed internally disordered polypeptides may play a critical role in cellular processes, cell signaling, and immunomodulation during viral infection [64,65,66].

## 4. Discussion

IAV populations are highly diverse swarms of genetically and biologically heterogeneous particle subpopulations. Understanding the factors contributing to the diversity of IAV subpopulations is crucial for developing future anti-influenza therapeutics and vaccines. DI particles typically contain one or more ID vRNA, with a large internal deletion that can render them replication-incompetent. DI particles emerge de novo during virus growth in host cells contingent upon viral and host-related factors [2,26,67], and DI particle abundance can fluctuate during viral passages [6,68,69,70]. However, the impact of IAV subculture and plaque purification on the ID vRNA abundance and diversity must be better understood. This study showed that the ID vRNA abundance within an IAV population can be maintained during a single virus subculture but not during the physical separation of viral subpopulations through plaque purification (Figure 2 and Figure 4, respectively).

Further, the current study provides direct evidence of relationship between the high IFN-inducing phenotype of IAV subpopulations and increased formation of DI particles [67] through enhanced generation of ID vRNAs with distinct deletion junctions (Figure 5B and Figure 6B). Finally, we uncovered a diverse population of in silico-predicted polypeptides that the ID vRNAs can encode (Figure 7 and Figure 8). Direct evidence is needed to determine the expression levels of these polypeptides and their roles in cellular processes and immunomodulation during infection.

Historically, the IAV DI particles were considered undesirable products of viral propagation at high multiplicities of infection [4,5]. We know that the multiplicity of infection does not solely control the generation of DI particles. Indeed, large numbers of DI particles often arise in viruses prepared under conditions that ensure low multiplicities of infection [6,10,67]. Therefore, it is unsurprising that numerous ID vRNAs with distinct deletion junctions were produced de novo in all clonal subpopulations isolated in this study (Figure 5). More than 90% of ID vRNAs originated from the polymerase genes, which corroborates the results of previous studies on the preferential generation of the ID vRNAs from the longest IAV genomic segments [7,10,22,23]. Despite significant differences observed in the distribution of ID vRNAs among the polymerase genes of the two PC4 virus preparations (Figure 2B), all polymerase genes tended to equally support the de novo generation of ID vRNAs as they emerged at similar levels from either of the three polymerase genes in the plaque-purified clones (Figure 5A,B).

Collectively, our results revealed a potential role for the NS1Δ76-86 mutation in the enhancement of the IFN responses and in increasing the likelihood of ID vRNA generation in the context of the PC4 virus (Figure 6B,C). Significant alterations in the constellations of genes or gene segments have been shown to trigger the generation of DI particle populations. For example, substituting the NS gene from a lethal H5N1 virus with that of an H1N1 virus has triggered the de novo generation of DI particles while enhancing the virus’s ability to induce IFNs [67]. More significant subpopulations of DI particles or ID vRNAs had also been reported in IAV that experienced genetic imbalances due to mutations in PA [24,71], nonstructural protein 2 [72], and matrix 1 and matrix 2 proteins [73]. The sudden increase in DI particle production may be related to incompatibility between the mutated genes and the genetic background of the virus [67,74] as well as instability of the viral RNA-polymerase complex [71,75,76,77]. Future investigations are needed to better understand the mechanisms governing the enhanced ID vRNA generation through the NS1Δ76-86 mutation, explicitly focusing on the interaction of NS1 with the viral replication/transcription complex [71,75,76,77,78] and the potential direct role of IFN in ID vRNA synthesis (e.g., during infection in the presence or absence of IFN stimulation).

Previous studies using IAVs enriched for DI particles that carry ID vRNAs revealed their ability to enhance IFN response in vitro [6], promote persistent infections in vitro [79], or reduce virulence in vivo [9], which were generally linked to the ability of ID vRNAs to outcompete the replication of the standard virus genome. However, investigations into the proteins encoded by ID vRNAs are lacking. Our in silico results suggest that the ID vRNAs may express a large population of truncated internally disordered proteins in the infected cells (Figure 7 and Figure 8). Such disordered polypeptides may exist as dynamic ensembles of rapidly interconverting conformations. Internally disordered proteins have garnered significant attention in molecular biology and biophysics because of their unique properties and essential roles in various cellular processes. They are highly versatile and can interact with various binding partners, including other proteins, nucleic acids, and small molecules. This versatility allows them to participate in numerous cellular processes and pathways, including molecular recognition, cell signaling and regulation, and immunomodulation [64,80,81].

Additionally, peptides liberated from such disordered proteins can preferentially bind to the major histocompatibility complex molecules, modulating the immunopeptidome toward metabolically unstable proteins [65]. To date, multiple mechanisms have been described for the formation of defective or non-canonical viral polypeptides in infected or stressed cells [66], but the potential of ID vRNA-derived polypeptides in immunomodulation, cellular processes, or immunity has yet to be uncovered [33,82,83,84]. We hypothesize that generating such non-canonical viral polypeptides is beneficial for virus survival in the hostile environment of the host. Further investigation is needed to explore the production of polypeptides derived from ID vRNAs and their potential involvement in cellular processes and immunity (e.g., in genetically engineered cells expressing defined ID vRNA-derived polypeptides). This includes investigating their role in binding to other ribonucleoproteins or modulating the immunopeptidome.

To our knowledge, this study is the first to report the whole genome profile of ID vRNAs within plaque-purified IAV subpopulations with different IFN-inducing capacities. Our results suggest that specific mutations within the IAV genome may enhance the de novo generation of ID vRNAs. Nevertheless, extremely diverse ID vRNAs appear to emerge and populate most IAV populations, which explains why we did not find a discriminative pattern in the parental or high IFN-inducing clones to distinguish them based on their ID vRNA junction profiles. Even so, the ID vRNAs investigated in this study were mainly derived from those successfully packaged into the DI particles, underestimating a potentially large population of intracellular ID vRNAs, including mini and small vRNAs, that fail to package into virions. Future studies should involve a broader range of IAVs with diverse IFN-inducing capacities, which are produced under various culture conditions. This approach may help elucidate the reason behind the higher presence of ID vRNAs in clones that induce higher IFN levels. Furthermore, conducting IAV infections in genetically engineered cell lines that express specific ID vRNAs could offer deeper insights into how defective viral genes relate to cellular processes and host immunity during virus evolution.

## Figures and Tables

**Figure 1 viruses-15-02107-f001:**
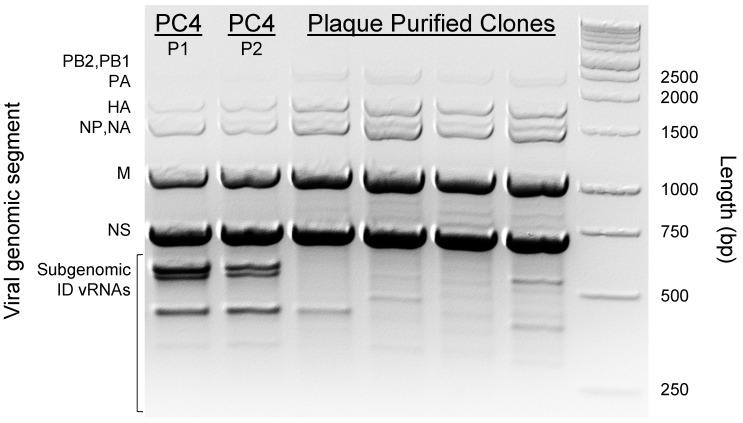
Whole genomic profile of the PC4 viruses and four plaque-purified viral clones. Full-length IAV genomes were amplified using a universal RT-PCR approach to amplify all IAV gene segments in a single reaction as described in Materials and Methods. The amplified products were visualized on a 2% *w*/*v* agarose gel stained with EtBr. The gel electrophoresis reaction was allowed to run for 105 min at 110 V in Tris-acetate-EDTA buffer. Subgenomic ID vRNAs in PC4 virus preparations appeared as three separate bands within ~250–700 bp size range below the truncated NS gene bands (700 bp). Abbreviations for IAV genome segments: PB2, polymerase basic 2; PB1, polymerase basic 1; PA, polymerase acidic; HA, hemagglutinin; NP, nucleoprotein; NA, neuraminidase; M, matrix; NS, nonstructural.

**Figure 2 viruses-15-02107-f002:**
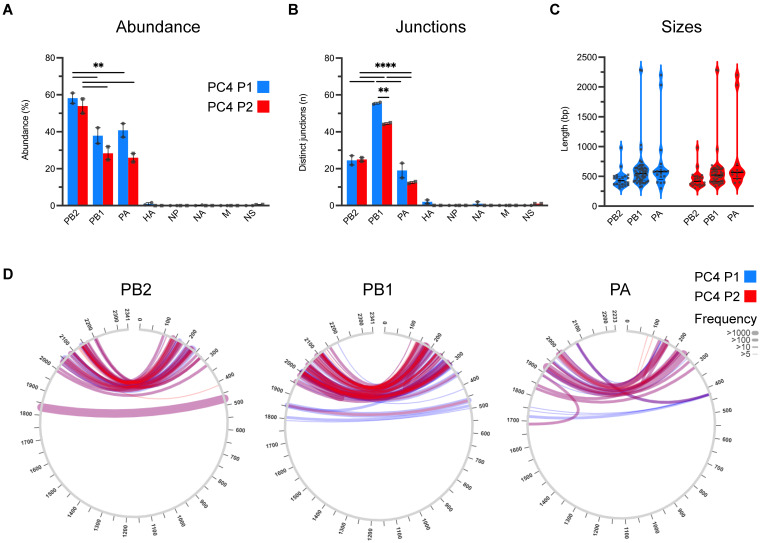
Abundance, junction count, size, and junction profiles of ID vRNAs identified in deep sequences obtained from the PC4 virus stocks (P1 in blue and P2 in red, respectively). (**A**) The abundance of ID vRNAs identified per genomic segment was determined by the cumulative percentage of sequencing reads corresponding to individual ID vRNAs from each gene over average sequence coverages at 5′ and 3′ termini of the gene, as described in Materials and Methods. (**B**) Number of distinct deletion junctions identified per segment. (**C**,**D**) Size (**C**) and junction (**D**) profiles of ID vRNAs originated from polymerase genes (PB2, PB1, and PA). The graphs and analyses included only the ID vRNAs with a percent abundance of ≥0.1%. Line thickness in panel (**D**) represents the frequency of reads corresponding to each deletion with minimum read support of ≥5. Dots represent values for individual samples. Bars indicate the mean ± SEM of groups. Statistical differences among the values derived from each gene between the two groups were evaluated using the unpaired *t*-test. Statistical differences among the values derived from different genes within each group were evaluated using the one-way ANOVA with Tukey’s post hoc test. For panels (**A**,**B**), the statistical differences are only shown for the deletions originating from polymerase genes (PB2, PB1, and PA) since they consistently showed significantly higher values than those derived from all the other genes. Asterisks above the line(s) indicate statistical differences between the groups (****, *p* < 0.0001; **, *p* < 0.01).

**Figure 3 viruses-15-02107-f003:**
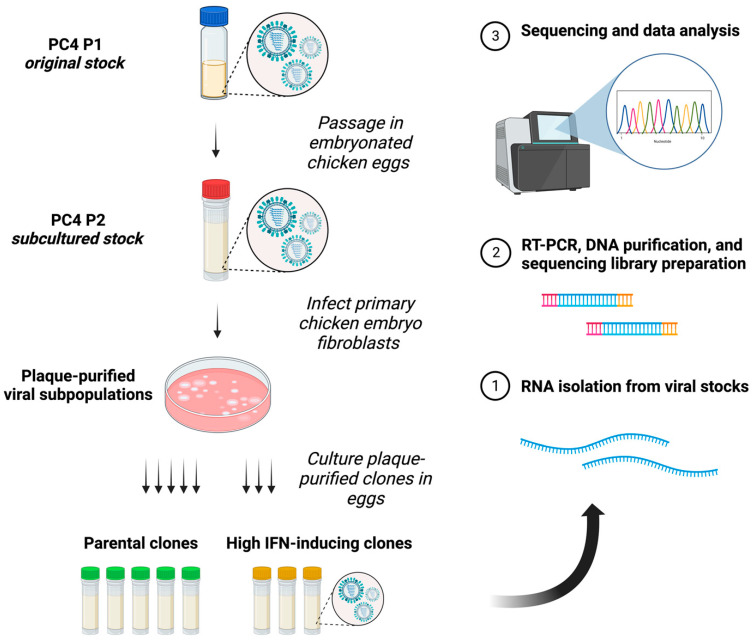
Schematic illustration of the experimental design and viral population hierarchy. Created with BioRender.com (accessed on 10 October 2023).

**Figure 4 viruses-15-02107-f004:**
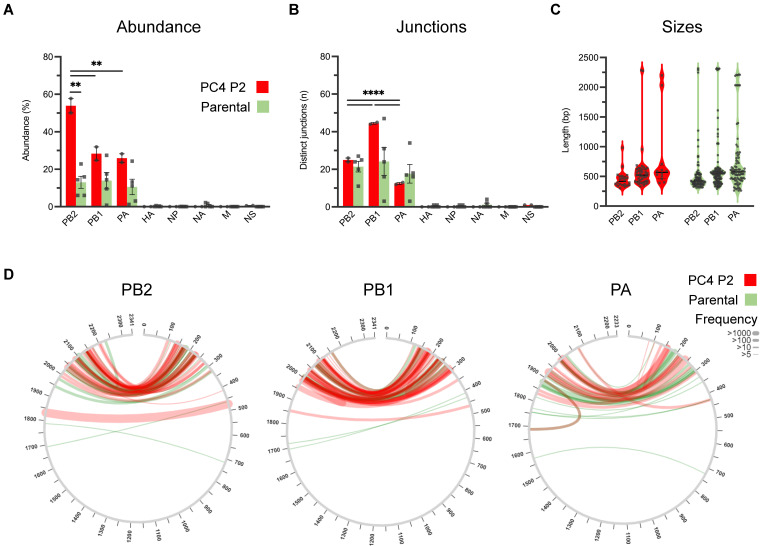
Abundance, junction count, size, and junction profiles of ID vRNAs identified in deep sequences obtained from the PC4 virus (in red) and its parental plaque-purified clones (in green). (**A**) The abundance of ID vRNAs identified per genomic segment. (**B**) Number of distinct deletion junctions identified per segment. (**C**,**D**) Size (**C**) and junction (**D**) profiles of ID vRNAs originated from polymerase genes (PB2, PB1, and PA). The graphs and analyses included only the ID vRNAs with a percent abundance of ≥0.1%. Line thickness in panel (**D**) represents the frequency of reads corresponding to each deletion with minimum read support of ≥5. Dots represent values for individual samples. Bars indicate the mean ± SEM of groups. Statistical differences among the values derived from each gene between the two groups were evaluated using the unpaired *t*-test. Statistical differences among the values derived from different genes within each group were evaluated using the one-way ANOVA with Tukey’s post hoc test. For panels (**A**,**B**), the statistical differences are only shown for the deletions originating from polymerase genes (PB2, PB1, and PA) since they consistently showed significantly higher values than those derived from all the other genes. Asterisks above the line(s) indicate statistical differences between the groups (****, *p* < 0.0001; **, *p* < 0.01).

**Figure 5 viruses-15-02107-f005:**
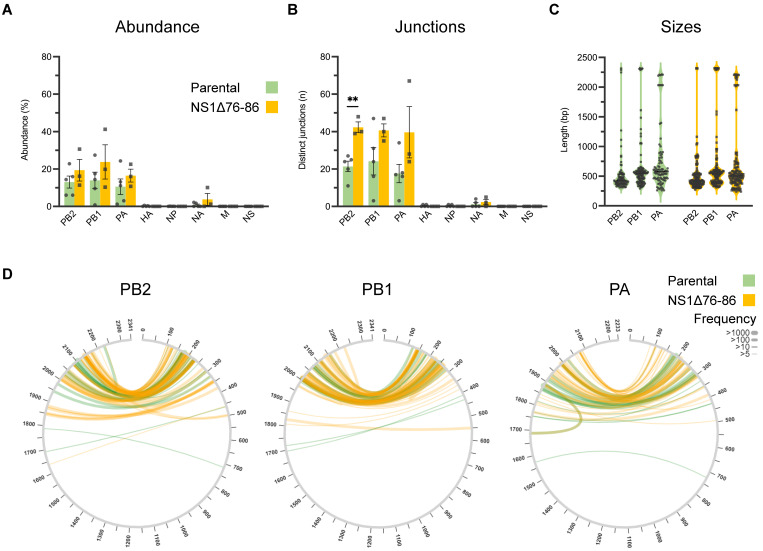
Abundance, junction count, size, and junction profiles of ID vRNAs identified in deep sequences obtained from the parental (in green) and high IFN-inducing clones possessing the NS1Δ76-86 deletion (in golden yellow). (**A**) The abundance of ID vRNAs identified per genomic segment. (**B**) Number of distinct deletion junctions identified per segment. (**C**,**D**) Size (**C**) and junction (**D**) profiles of ID vRNAs originated from polymerase genes (PB2, PB1, and PA). The graphs and analyses included only the ID vRNAs with a percent abundance of ≥0.1%. Line thickness in panel (**D**) represents the frequency of reads corresponding to each deletion with minimum read support of ≥5. Dots represent values for individual samples. Bars indicate the mean ± SEM of groups. Statistical differences among the values derived from each gene between the two groups were evaluated using the unpaired *t*-test. Statistical differences among the values derived from different genes within each group were evaluated using the one-way ANOVA with Tukey’s post hoc test. Asterisks indicate statistical differences (**, *p* < 0.01).

**Figure 6 viruses-15-02107-f006:**
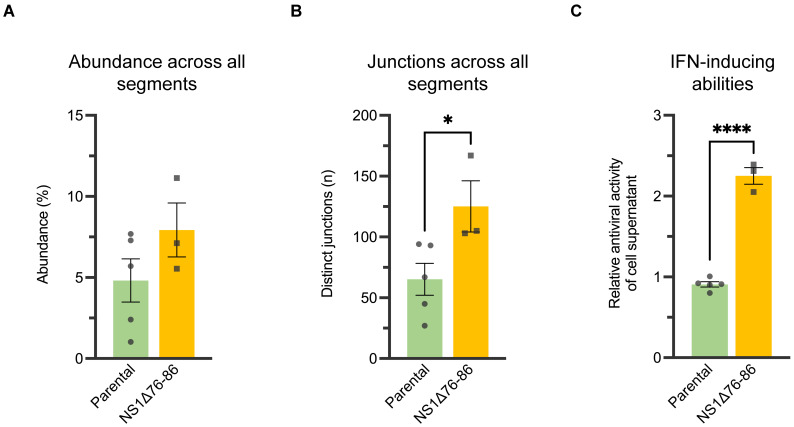
The genome-wide abundance of ID vRNAs and their distinct junctions in relation to the IFN-inducing ability of the parental (in green) versus high IFN-inducing clones possessing the NS1Δ76-86 deletion (in golden yellow). (**A**) Total abundance of ID vRNAs identified across all genomic segments. (**B**) Total number of distinct deletion junctions identified across all genomic segments. (**C**) Relative IFN-inducing capacities of the plaque-purified clones in cell cultures. IFN induction (MOI = 0.3) and bioassay were performed as described in Materials and Methods. The graphs and analyses included only the ID vRNAs with a percent abundance of ≥0.1%. Dots represent values for individual samples. Bars indicate the mean ± SEM of groups. Statistical differences between the groups were evaluated using the unpaired *t*-test. Asterisks indicate statistical differences (****, *p* < 0.0001; *, *p* < 0.05).

**Figure 7 viruses-15-02107-f007:**
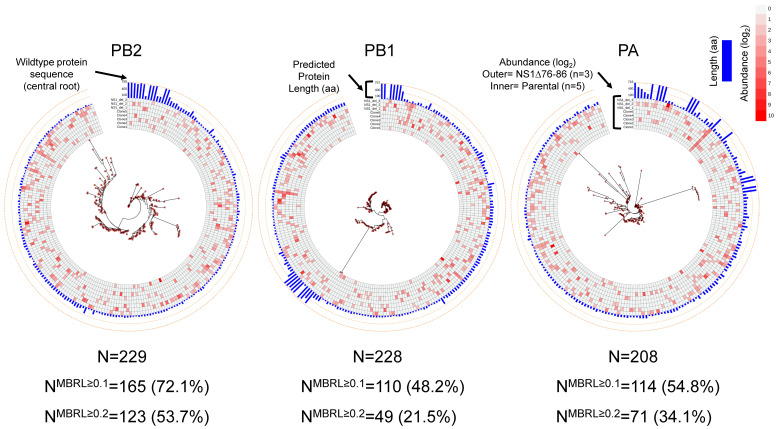
Extreme phylogenetic diversity of putative non-standard polypeptides generated from the ID vRNAs identified in the PB2, PB1, and PA genes. Multiple sequence alignments of the putative polypeptide sequences were performed based on the Clustal W method on slow/accurate mode and BLOSUM weight matrix on the GenomeNet (https://www.genome.jp/tools-bin/clustalw) [47,48,49]. Phylogenetic reconstructions were performed using the function “build” of ETE3 3.1.2 [50] as implemented on the GenomeNet (https://www.genome.jp/tools/ete/). The trees were constructed using FastTree with slow NNI and MLACC = 3 (to make the maximum-likelihood NNIs more exhaustive) [51]. Phylogenetic trees and the additional metadata for ID vRNA abundance and protein lengths were visualized using the Interactive Tree Of Life (iTOL) [57]. Branch lengths represent the evolutionary distances computed using the Poisson correction method based on the number of amino acid substitutions per site [48]. The analyses involved 229, 228, and 208 amino acid sequences or leaf nodes (N) for the PB2, PB1, and PA genes, respectively. NMBRL ≥ 0.1 and NMBRL ≥ 0.2 represent the number of visible leaf nodes after collapsing all the clades whose mean branch length distance to their leaves was below 0.1 and 0.2, respectively. Red dots represent the nodes. The root node at the center of each tree corresponds to the standard amino acid sequences of the designated viral gene.

**Figure 8 viruses-15-02107-f008:**
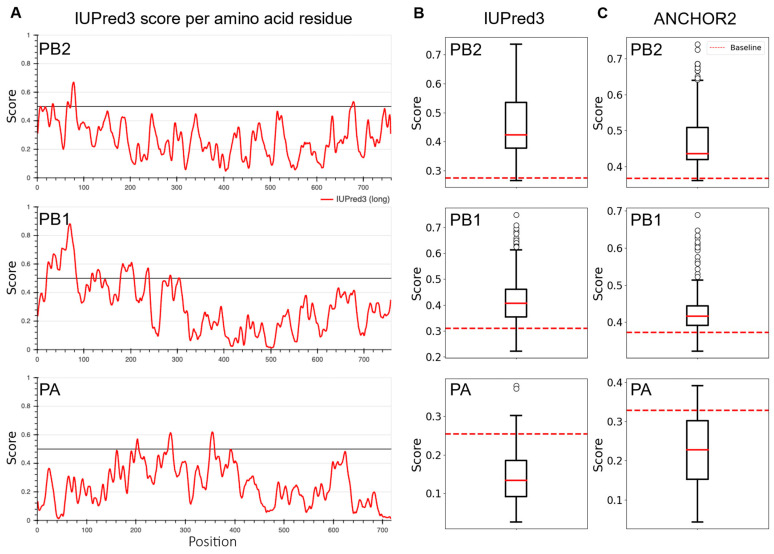
Disordered nature of putative viral polypeptides encoded by the ID vRNAs. Prediction of intrinsically disordered and protein binding regions was performed using the IUPred3 and ANCHOR2 algorithms with the default “long disorder” option [52,53,54]. (**A**) The red lines represent the disorder prediction score of each amino acid residue for intact PB2, PB1, and PA (probability values from ‘0’, least likely, to ‘1’, most likely). The predicted model accurately reflects the structured nature of the PB2 cap-binding domain and PB1 active site (in the middle), and PA endonuclease domain (at the N-terminal) based on the decreased tendency for disorder. The N-terminal and C-terminal binding motifs of PB2 and PB1 exhibit a higher tendency for disorder, a common trait found in disordered binding sites that interact with other proteins. Average IUPred3 (**B**) and ANCHOR2 (**C**) values for proposed polypeptides from ID vRNAs identified in the plaque-purified clones. Red dotted lines represent the mean value estimated for the intact PB2, PB1, or PA viral proteins.

## Data Availability

Any supplementary data not published within the context of this article will be provided upon request by corresponding author.

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
