# Peer review of "Diversity and Complexity of Internally Deleted Viral Genomes in Influenza A Virus Subpopulations with Enhanced Interferon-Inducing Phenotypes"

_viruses, 2023, doi:10.3390/v15102107_

Round 1
Reviewer 1 Report
In their manuscript Ghorbani and co-authors attempt to describe the complexity and diversity of internally truncated defective viral genomes of a panel of avian influenza viruses with truncated NS1 proteins and high IFN stimulatory characteristics. Using deep sequencing of virus stocks they identified - despite an ability to still generate ID vRNA de novo - a loss ID vRNA abundance and complexity both after a single passaging step and after plaque purification of virus clones. Furthermore, they observed that clones containing additional truncations in the NS1 protein appeared to induce higher IFN responses and more frequent unique ID vRNA junctions without generating more total ID vRNA compared to the parental clones. Finally, the authors determine using an in silico approach that ID vRNAs hypothetically can encode many novel polypeptides that on average would be significantly more disordered than their respective wild-type counterparts. They speculate that these polypeptides may play an important role in cell processes and signalling pathways, affecting viral infections and the host response.
The manuscript heavily relies on and refers to previous related publications by the authors. Despite some neat findings, the authors struggle to convey the importance or significance of their findings to the reader and in fact, their most interesting findings remain speculative.
Comments:
Line 203: Since RNA is extracted using the QIAamp Viral RNA Mini kit any RNA smaller than 200 nt will be lost. It is known for IAV that specific small ID vRNAs (mvRNA) are highly immunogenic and strongly correlate with IFN stimulation. Nevertheless, mvRNAs are entirely disregarded in this analysis here and are not even commented on.
Line 227: The authors write "relative abundance of ID vRNA per nucleotide of the viral genome". It is unclear from the figure and figure description what 'per nucleotide' means here or whether 'per segment' was intended here.
Line 236: Loss of larger sized ID vRNAs for the PB1 segment as described in the text can not be appreciated or seen in panel 2C. Either the panel is too unclear/small or the text does not match the shown data.
Fig. 2A-B and 3A-B: Are we to understand that the asterisks of significance in the graphs are valid for all lines underneath? This is not stated anywhere.
Line 262: Here (and in other passages later) the authors mention PC4 P2 and 5 parental clones (I assume from a previous publication). It is however unclear whether these 5 clones are plaque purified clones of the P1 population or whether they are clones of the PC4 P2 population (which is mentioned to be used for plaque purification of clones) and confusingly then called 'parental' clones. Either the language/description of the different populations and clones should be refined to make it clear to the reader what the relationship is between the different compared viruses or a schematic could be introduced, e.g. in Figure 1.
Line 333: total number of "unique" deletion junctions.
Line 335: Figure 5B is mentioned before 5A.
Line 340: The authors state that it is unclear whether these ID vRNAs play a role in the higher IFN response or whether their abundance is a by-product of a higher IFN response. It would be an easy experiment to test at least the latter hypothesis: infect with the same virus stock in the presence or absence of IFN stimulation and compare the resulting ID vRNA population.
Line 402: It would be interesting and strongly strengthen the manuscript if some / the most abundant or conserved of these polypeptides are cloned and expressed in cells to see their effect both in the absence or presence of viral infections.
Line 444: longest IAV genomic segments (not significant)
In some passages (see previous comments) I suspect that the use of the language is probably not entirely accurate or precise, making it difficult to understand some details of the text.
Author Response
Reviewer #1 (Comments for the Author):
In their manuscript Ghorbani and co-authors attempt to describe the complexity and diversity of internally truncated defective viral genomes of a panel of avian influenza viruses with truncated NS1 proteins and high IFN stimulatory characteristics. Using deep sequencing of virus stocks they identified - despite an ability to still generate ID vRNA de novo - a loss ID vRNA abundance and complexity both after a single passaging step and after plaque purification of virus clones. Furthermore, they observed that clones containing additional truncations in the NS1 protein appeared to induce higher IFN responses and more frequent unique ID vRNA junctions without generating more total ID vRNA compared to the parental clones. Finally, the authors determine using an in silico approach that ID vRNAs hypothetically can encode many novel polypeptides that on average would be significantly more disordered than their respective wild-type counterparts. They speculate that these polypeptides may play an important role in cell processes and signaling pathways, affecting viral infections and the host response.
The manuscript heavily relies on and refers to previous related publications by the authors. Despite some neat findings, the authors struggle to convey the importance or significance of their findings to the reader and in fact, their most interesting findings remain speculative.
Response:
We are grateful that the reviewer appreciates our efforts to describe the complexity and diversity of internally truncated defective viral genomes within a panel of NS1-truncated IAVs. As the reviewer pointed out, while more work is needed to carefully examine the mechanisms involved with the observed enhanced generation of unique ID vRNA junctions in the high IFN-inducing clones in the future, we believe that the reviewer’s suggestions have led to significant improvement of the revised manuscript. Accordingly, we further emphasized and suggested additional experiments as future directions, which is explained in the point-by-point responses to the reviewer’s comments below.
Comments:
Line 203: Since RNA is extracted using the QIAamp Viral RNA Mini kit any RNA smaller than 200 nt will be lost. It is known for IAV that specific small ID vRNAs (mvRNA) are highly immunogenic and strongly correlate with IFN stimulation. Nevertheless, mvRNAs are entirely disregarded in this analysis here and are not even commented on.
Response:
As the reviewer pointed out, our data set do not contain the smaller sized ID vRNAs including mini and small vRNAs. We added a statement within the Introduction (lines 55-59) in the revised manuscript acknowledging the presence of these smaller sized ID vRNAs (“In addition to DI particle-derived ID vRNAs, which are typically larger than ~200 nucleotides, there are other classes of short aberrant vRNAs, namely mini and small vRNAs, with lengths of ~50-125 and 22-27 nucleotides, respectively, which are less likely to be encapsulated within influenza virions [27, 28]”). In addition to this, we added another statement within the Introduction section (lines 84-85) to further emphasize that the purpose of the study was to analyze the larger “DI particle-derived ID vRNAs that can be packaged within the virions (“…characterize the DI particle-derived ID vRNAs within viral populations”). Furthermore, we also emphasized on these facts by adding the following statement within the Conclusion section (line 515) (“…large population of intracellular ID vRNAs, including mini and small vRNAs, that fail to package into virions…”).
Accordingly, the following references were added to the References section regarding mini and small vRNAs:
- te Velthuis, A. J.; Long, J. C.; Bauer, D. L.; Fan, R. L.; Yen, H.-L.; Sharps, J.; Siegers, J. Y.; Killip, M. J.; French, H.; Oliva-Martín, M. J., Mini viral RNAs act as innate immune agonists during influenza virus infection. Nat. Microbiol. 2018, 3, (11), 1234-1242.
- Fodor, E.; Te Velthuis, A. J., Structure and function of the influenza virus transcription and replication machinery. Cold Spring Harbor perspectives in medicine 2020, 10, (9). a038398.
- Perez, J. T.; Varble, A.; Sachidanandam, R.; Zlatev, I.; Manoharan, M.; García-Sastre, A.; tenOever, B. R., Influenza A virus-generated small RNAs regulate the switch from transcription to replication. Proceedings of the National Academy of Sciences 2010, 107, (25), 11525-11530.
- Koire, A.; Gilbert, B. E.; Sucgang, R.; Zechiedrich, L., Repurposed transcriptomic data reveal small viral RNA produced by influenza virus during infection in mice. PloS one 2016, 11, (10), e0165729.
Line 227: The authors write "relative abundance of ID vRNA per nucleotide of the viral genome". It is unclear from the figure and figure description what 'per nucleotide' means here or whether 'per segment' was intended here.
Response:
Thank you for pointing out this issue. To avoid further confusions, we modified this statement (lines 235-237) by adding the following additional information in parenthesis “relative abundance of ID vRNAs per nucleotide of each viral genome segment (relative abundance divided by segment size)” to reflect on the fact this statement refers to relative number of ID vRNAs normalized by segment size. To further clarify that this statement does not directly refer to the manuscript figures, we also added “data not shown” at the end of the sentence. As the reviewer pointed out and mentioned in the figure legends, the relative numbers presented in the manuscript figures are not normalized per nucleotide of the segment and plotted as numbers per segment.
Line 236: Loss of larger sized ID vRNAs for the PB1 segment as described in the text can not be appreciated or seen in panel 2C. Either the panel is too unclear/small or the text does not match the shown data.
Response:
Thank you for pointing out this issue. Here we are referring to a few visible lines that are lost between 500 and 1800 bp for PB1 in Figure 2D or dots that are lost within 1000 bp range in Figure 2C. Also, we understand that this is not solid evidence for the observed reduction in the ID vRNA. Therefore, we modified the statement (lines 243-245) by adding additional explanation and toning down this statement as followed (“…which might be related to the loss of some larger-sized ID vRNAs (Figure 2B, C (e.g., length ~1000 bp), and D (e.g., break and rejoin sites at ~500 and 1800 bp, respectively)”)
Fig. 2A-B and 3A-B: Are we to understand that the asterisks of significance in the graphs are valid for all lines underneath? This is not stated anywhere.
Response:
We appreciate the reviewer for expressing his/her concern about the visualization of the statistical inferences. We modified the figure legends for figures 2 and 4 (lines 268 and 312-313) to indicate that “Asterisks above the line(s) indicate statistical differences between the groups.”
Line 262: Here (and in other passages later) the authors mention PC4 P2 and 5 parental clones (I assume from a previous publication). It is however unclear whether these 5 clones are plaque purified clones of the P1 population or whether they are clones of the PC4 P2 population (which is mentioned to be used for plaque purification of clones) and confusingly then called 'parental' clones. Either the language/description of the different populations and clones should be refined to make it clear to the reader what the relationship is between the different compared viruses or a schematic could be introduced, e.g. in Figure 1.
Response:
To address this issue and avoid any confusion for the readers, we created a schematic illustration (Figure 3, lines 294-297) according to the reviewer’s suggestion. Accordingly, we added a statement referring to Figure 3 between lines 275-277 (“Figure 3 offers a schematic illustration of the experimental design and population hierarchy pertaining to PC4-related viral subpopulations”). All subsequent figures were renamed. It is noteworthy that parental clones had also been defined earlier in line 79-80 (“the majority of viral subpopulations with similar genotypes/phenotypes (referred to as parental clones)”).
Line 333: total number of "unique" deletion junctions.
Response:
Revised accordingly (line 351).
Line 335: Figure 5B is mentioned before 5A.
Response:
This issue was resolved by revising the sentence between lines 349-353 (“While the total abundance of ID vRNAs was not statistically different between the parental and NS1Δ76-86 clones (Figure 6A), the total number of unique deletion junctions detected from all eight genomic segments combined was also significantly higher in the NS1Δ76-86 clones than in the parental clones (Figure 6B)”)
Line 340: The authors state that it is unclear whether these ID vRNAs play a role in the higher IFN response or whether their abundance is a by-product of a higher IFN response. It would be an easy experiment to test at least the latter hypothesis: infect with the same virus stock in the presence or absence of IFN stimulation and compare the resulting ID vRNA population.
Response:
We appreciate the comments pointing out the limitation of our study and acknowledge that additional studies are needed in future. We made some necessary changes within the Discussion section in the revised manuscript to address the reviewer’s concerns with regards to suggested future experiment in line 480: “(e.g., during infection in the presence or absence of IFN stimulation)”. However, we respectfully disagree that the suggested experiment and following analysis is minor work and, more importantly, we believe it is better to be addressed together with other limitations of the current study with focus on better understanding the mechanism in our future studies.
Line 402: It would be interesting and strongly strengthen the manuscript if some / the most abundant or conserved of these polypeptides are cloned and expressed in cells to see their effect both in the absence or presence of viral infections.
Response:
Again, we really appreciate the comments pointing out the limitation of our study and acknowledge that additional studies are needed in future. Similar to our answer on the previous comment, we made additional changes in the Discussion section by adding the following statement between lines 504-505: “(e.g., in genetically engineered cells expressing defined ID vRNA-derived polypeptides).”.
Please also note that we have proposed similar experiments in our concluding remarks (lines 516-522) (“Future studies should involve a broader range of IAVs with diverse IFN-inducing capacities, which are produced under various culture conditions. This approach may help elucidate the reason behind the higher presence of ID vRNAs in clones that induce higher IFN levels. Furthermore, conducting IAV infections in genetically engineered cell lines that express specific ID vRNAs could offer deeper insights into how defective viral genes relate to cellular processes and host immunity during virus evolution”)
Line 444: longest IAV genomic segments (not significant)
Response:
Revised accordingly (line 460).
Reviewer 2 Report
Scientific articles are well collected in the "Introduction" (a large number of references). The results of the study expand our knowledge of influenza virus ID-vRNAs and the truncated proteins they encode.
I recommend increasing the quality or size of the Figures in the Supplementary. Especially S.Fig. 2 - the legend, the labels of the axes of each graph are not visible at all.
Author Response
Reviewer #2 (Comments for the Author):
Scientific articles are well collected in the "Introduction" (a large number of references). The results of the study expand our knowledge of influenza virus ID-vRNAs and the truncated proteins they encode.
I recommend increasing the quality or size of the Figures in the Supplementary. Especially S.Fig. 2 - the legend, the labels of the axes of each graph are not visible at all.
Response:
Many thanks to the reviewer for acknowledging the value of our study in expanding our knowledge of IAV ID vRNAs and the potential truncated proteins they may encode. We appreciate the important points raised by the reviewer and made sure to enhance the main and supplementary figures accordingly in the revised manuscript.
Reviewer 3 Report
This is a very important, careful, and nice study. Only two minor suggestions.
Line 61-62: “We have previously introduced the plaque clone 4 (PC4) virus, with a severely truncated nonstructural 1 (NS1) protein effector domain…”. It will be better if the background information of the NS1 normal function is followed this sentence, as the truncated NS1 is a focus of the study.
Line 431-432: “… with distinct deletion junctions (Figure 4B and 17B).” Check this.
Author Response
Reviewer #3 (Comments for the Author):
This is a very important, careful, and nice study. Only two minor suggestions.
Line 61-62: “We have previously introduced the plaque clone 4 (PC4) virus, with a severely truncated nonstructural 1 (NS1) protein effector domain…”. It will be better if the background information of the NS1 normal function is followed this sentence, as the truncated NS1 is a focus of the study.
Response:
We deeply appreciate the reviewer’s comment on the importance and thoroughness of our experimental design. Thank you for your excellent suggestion. To address the reviewer’s suggestion, we revised the manuscript to provide the following additional information about influenza NS1 protein and its importance in designing live attenuated vaccines between lines 67-70 (“The NS1 protein serves a pivotal role in antagonizing the host's IFN response, and its truncation in IAVs results in enhanced innate immune responses, making it a promising candidate for the development of highly attenuated and effective LAIVs [39, 40]”)
Accordingly, the following references were added to the References section:
- Rosário-Ferreira, N.; Preto, A. J.; Melo, R.; Moreira, I. S.; Brito, R. M., The Central Role of Non-Structural Protein 1 (NS1) in Influenza Biology and Infection. Int J Mol Sci 2020, 21, (4), 1511.
- Richt, J. A.; García-Sastre, A., Attenuated influenza virus vaccines with modified NS1 proteins. In Vaccines for Pandemic Influenza, Compans, R. W.; Orenstein, W. A., Eds. Springer-Verlag: Heidelberg, Germany, 2009; pp 177-195.
Line 431-432: “… with distinct deletion junctions (Figure 4B and 17B).” Check this.
Response:
Thank you for pointing out this issue. We fixed this typo between lines 447-448 in the revised manuscript (Figure 5B and 6B).